# Associations between Dietary Patterns and Metabolic Syndrome: Findings of the Korean National Health and Nutrition Examination Survey

**DOI:** 10.3390/nu15122676

**Published:** 2023-06-08

**Authors:** Yun-Ah Lee, Sang-Wook Song, Se-Hong Kim, Ha-Na Kim

**Affiliations:** Department of Family Medicine, St. Vincent’s Hospital, College of Medicine, The Catholic University of Korea, Seoul 06591, Republic of Korea

**Keywords:** metabolic syndrome, dietary patterns, carbohydrate, fat, protein

## Abstract

Metabolic syndrome (MetS) is a multifactorial cluster of metabolic disorders related to cardiovascular disease and type 2 diabetes mellitus. Diet and dietary patterns are significant factors in the development and management of MetS. The associations between dietary patterns (i.e., high-carbohydrate [HCHO], high-fat [HF], and high-protein [HP] diets) and the prevalence of MetS in Koreans were examined using data from the Korean National Health and Nutrition Examination Survey, collected between 2018 and 2020. The study included data from 9069 participants (3777 men and 5292 women). The percentage of participants with MetS was significantly higher in the HCHO diet group than in the normal diet group in women. Women with HCHO diet were positively associated with elevated blood pressure and triglyceride levels based on a comparison with the normal diet group (*p* = 0.032 and *p* = 0.005, respectively). Men with an HF diet were negatively associated with elevated fasting glucose levels based on a comparison with the normal diet group (*p* = 0.014). Our findings showed that HCHO intake was strongly associated with a higher risk of MetS, especially elevated blood pressure and triglyceride levels in women, and an HF diet was negatively associated with elevated fasting glucose levels in men. Further prospective studies of the impact of dietary carbohydrate, fat, and protein proportions on metabolic health are needed. The optimal types and proportions of these dietary components, as well as the underlying mechanisms through which suboptimal proportions can lead to MetS, should also be investigated.

## 1. Introduction

Metabolic syndrome (MetS) comprises a group of metabolic abnormalities including hyperglycemia, raised blood pressure, abdominal obesity, low high-density lipoprotein cholesterol (HDL-C), and hypertriglyceridemia. MetS is associated with an increased risk of type 2 diabetes mellitus and cardiovascular disease (CVD) [1,2,3,4], both of which have increased in prevalence and are now global health problems that impose large economic burdens on public health systems [5,6,7]. The prevalence of MetS has also increased worldwide, with a gradual increase occurring in Korea over the past decade [8,9,10,11]. The prevalence of MetS in Korea was 28.1% for men and 18.7% for women as of 2017 [12]. MetS has caused serious health problems in children and adolescents as well [13]. The prevalence of MetS in young people varies according to the criteria used: 8.9, 15.6, 34.9, and 8.9%, according to definitions suggested by IDF [14], Cook [15], and de Ferranti [16], respectively. MetS can be affected by age, sex, metabolism, genetics, and environmental factors, such as drinking, smoking, diet, and physical inactivity [17,18]. Diet and dietary patterns are important with respect to the development and management of MetS and its risk factors [19,20,21,22,23]. Dietary quality and patterns have been associated with MetS [24]. A systematic review showed that an increased consumption of saturated fatty acids is related to an increased risk of MetS [25]. A previous meta-analysis revealed that high intake of monounsaturated fatty acids is associated with lower blood pressures based on a comparison with the high carbohydrate, low fat diet group [26]. However, previous studies have reported inconsistent findings regarding the association between the risk of MetS and dietary macronutrient intake [25,27,28,29,30]. Among women with a body mass index (BMI) of ≥25 kg/m^2^, the risk of MetS was significantly higher in the highest than the lowest quintile of carbohydrate intake. However, this was not seen for women with a BMI < 25 kg/m^2^ [28]. Low HDL-C is associated with high carbohydrate intake, regardless of energy or fat intake [27]. The risk of MetS significantly increased in men with higher percentages of total energy intake from carbohydrates, regardless of fat intake. In women, the risk of MetS significantly increased only among those with very high percentages of carbohydrate intake and very low percentages of fat intake [30]. However, few studies have evaluated the associations between the patterns of macronutrient intakes (e.g., high carbohydrate, high fat, and high protein) and MetS according to sex in the Korean population. It is not yet clear whether a high-carbohydrate (HCHO) diet or a high-fat (HF) diet or a high-protein (HP) diet is metabolically more deleterious. Therefore, in this study, data from the Korean National Health and Nutrition Examination Survey (KNHANES) were analyzed to investigate the potential associations of carbohydrate, protein, and fat intakes with MetS in Korean adults.

## 2. Materials and Methods

### 2.1. Study Population

The data analyzed in this study were obtained from the KNHANES and collected between 2018 and 2020. The KNHANES is conducted by the Korean Center for Disease Control and Prevention at 3-year intervals to evaluate public health and provide baseline data to assess, develop, and establish public health policies for the Korean population. KNHANES participants are ≥1 year of age and non-institutionalized; they are chosen using a stratified, multi-stage cluster probability sampling method to ensure a nationally representative, independent, and homogeneous sample. Data are collected via household interviews and anthropometric, biochemical, and nutritional status assessments [31]. All protocols were approved by the Institutional Review Board of the Korean Center for Disease Control and Prevention, and all participants provided written informed consent at baseline.

In the KNHANES 2018−2020, 31,261 participants were recruited, of whom 23,461 completed the survey (participation rate: 75.0%). In the present cross-sectional study, the data from 19,038 adults aged ≥20 years were examined. Participants with missing data for major variables (*n* = 1228) were excluded, along with those who reported implausible energy intakes (<500 or >5000 kcal/day; *n* = 3180), had intakes on the day of the 24-h dietary recall survey that were not representative of their usual intakes (*n* = 4285), or did not meet all dietary criteria (*n* = 1276). Thus, the data of 9069 participants were finally examined in this study (Figure 1), which was approved by the Institutional Review Board (IRB) of the Catholic University of Korea (IRB approval number: VC19ZOSI0071).

### 2.2. Dietary Assessments

Trained interviewers estimated the dietary intakes of the participants using the 24-h dietary recall method, including all foods and beverages consumed in the previous 24 h. The nutrition survey was performed at the participants’ homes, and additional tools such as two-dimensional food volumes, containers, and food models were used to help participants recall their nutrient intakes. Dietary intake was estimated from the food composition tables of the Rural Development Administration, in combination with the nutrient database of the Korea Health and Industry of Development Institute [32,33]. An HCHO diet was defined as >65% carbohydrate, ≤30% fat, and ≤20% protein, in terms of the proportions of total energy. An HF diet was defined as ≤65% carbohydrate, >30% fat, and ≤20% protein, and an HP diet as ≤65% carbohydrate, ≤30% fat, and >20% protein. The energy derived from carbohydrate, fat, and protein intake was calculated based on standard conversion factors used to convert grams to kilocalories (4 kcal/g for carbohydrate, 9 kcal/g for fat, and 4 kcal/g for protein), and the energy derived from each of the three macronutrients as a proportion of total energy intake (%) was calculated. The normal diet group included participants who met the Dietary Reference Intakes for Koreans (55–65% carbohydrate, 15–30% fat, and 7–20% protein) [33].

### 2.3. Definition of Metabolic Syndrome

MetS was defined using the revised criteria of the National Cholesterol Education Program Adult Treatment Panel III (NCEP-ATP III) [34], i.e., by the presence of three or more of the following: waist circumference ≥ 90 cm for males and ≥85 cm for females [35]; triglyceride level ≥ 150 mg/dL or on medication to reduce the triglyceride level; high-density lipoprotein (HDL) cholesterol level < 40 mg/dL for males and <50 mg/dL for females or on medication to reduce HDL-cholesterol levels; systolic blood pressure ≥ 130 mmHg or diastolic blood pressure ≥ 85 mmHg, or on antihypertensive medication; and fasting glucose ≥ 100 mg/dL or on medication to reduce the glucose level.

### 2.4. Other Variables

Self-reported age, sex, smoking status, alcohol consumption, aerobic exercise, and household income data were obtained. Cigarette smoking was categorized based on current use estimates. Information on alcohol consumption included the frequency of drinking and number of drinks consumed per drinking day during the 1 year that preceded the household interview for KNHANES. The Korean criterion for a “standard drink” (any drink containing 10 g of pure alcohol) was used, assuming 4.5 vol% for beer, 12 vol% for wine, 6 vol% for makgeolli (a traditional Korean beverage), 21 vol% for Korean soju, and 40 vol% for whisky. Heavy drinking was defined as >14 standard drinks per week for men and >7 for women [36]. Self-reported physical activity undertaken during the week before the interview was assessed. Aerobic exercise was classified as “no” or “yes”, with participants not undertaking aerobic exercise included in the “no aerobic exercise” group. Household income was classified according to monthly equivalized household income quartiles, estimated as total monthly household income divided by the square root of the total number of household members. Body weight, height, and waist circumference were measured after an overnight fast with the participants wearing light indoor clothing without shoes. Waist circumference was measured using a tape measure around the umbilical region after exhalation, in the horizontal plane. BMI was calculated as weight (kilograms) divided by height (meters) squared.

### 2.5. Statistical Analysis

The SAS PROC SURVEY module, which considers strata, clusters, and weights, was used to analyze the data according to a complex sampling design. All analyses were performed using the sample weights from KNHANES. The sex-specific characteristics of the study population were analyzed using independent *t*-tests for continuous variables and a chi-squared test for dichotomous variables. Data are expressed as mean ± standard error, or as a percentage. Multiple logistic regression analysis was used to examine the associations between dietary patterns and MetS components after adjusting for covariates. Model 1 was adjusted for age; model 2 for age, smoking, alcohol consumption, BMI, physical activity, and household income; and model 3 for age, smoking, alcohol consumption, BMI, physical activity, household income, and total energy and fiber intakes. The percentages of MetS and number of MetS components in each group were analyzed using the complex sample Rao–Scott adjusted chi-square test. All statistical analyses were performed using SAS software (ver. 9.2; SAS Institute, Cary, NC, USA). *p*-values < 0.05 were considered significant.

## 3. Results

### 3.1. Characteristics of the Study Participants According to Dietary Patterns

The study included data from 9069 participants (3777 men and 5292 women) with a mean age of 51.01 ± 0.31 years (men: 49.81 ± 0.40 years, women: 52.13 ± 0.33 years). The prevalence rates of normal, HCHO, HF, and HP diets were 25.9%, 55.2%, 10.8%, and 8.1%, respectively, in men and 22.8%, 59.6%, 11.7%, and 5.9%, respectively, in women. Significant differences in age, smoking status, alcohol consumption, aerobic activity adherence, energy intake, carbohydrate intake, fat intake, protein intake, histories of diabetes, hypertension, and dyslipidemia according to dietary patterns were seen in both men and women (Table 1).

### 3.2. Percentages of Participants with Metabolic Syndrome According to Dietary Pattern

Figure 2 shows the percentages of MetS in each group after adjusting for covariates. The percentage of participants with MetS was significantly higher in the HCHO diet group than in the normal diet group in women (*p* < 0.0001) (Figure 2).

### 3.3. Associations between Metabolic Syndrome Components and Dietary Patterns

The unadjusted odds ratios (ORs), age-adjusted ORs (model 1), and multivariate-adjusted ORs (models 2 and 3) of MetS components according to dietary patterns and sex are shown in Table 2. In men, an HF diet was negatively associated with elevated fasting glucose levels based on a comparison with the normal diet group (OR = 0.579, 95% CI: 0.433–0.774, *p* < 0.001). This negative association remained significant after adjusting for age, BMI, smoking status, alcohol consumption, aerobic activity adherence, household income, daily energy intake, and fiber intake (OR = 0.671, 95% CI: 0.488–0.922, *p* = 0.014). In women, an HCHO diet was positively associated with elevated blood pressure and triglycerides levels based on a comparison with the normal diet group (OR = 2.213, 95% CI: 1.873–2.615, *p* < 0.001; OR = 1.902, 95% CI: 1.609–2.248, *p* < 0.001, respectively). These associations remained significant after adjusting for age, BMI, smoking status, alcohol consumption, aerobic activity adherence, household income, daily energy intake, and fiber intake (OR = 1.257, 95% CI: 1.021–1.548, *p* = 0.032; OR = 1.329, 95% CI: 1.091–1.619, *p* = 0.005, respectively) (Table 2).

### 3.4. Percentage of Participants with MetS in Each Diet Group According to the Number of Metabolic Syndrome Components

Figure 3 shows the weighted percentages of participants with MetS in the normal, HCHO, HF, and HP diet groups according to the number of MetS components. For both men and women, significant differences (*p* < 0.001) in the number of MetS components were seen among the diet groups. For both sexes, as the number of MetS components increased, so too did the percentage of participants with MetS; the percentage was higher in the HCHO group than in the normal diet, HF, and HP groups (*p* < 0.001), and was also lower in the HF group than in the normal diet group (men, *p* = 0.029; women, *p* = 0.033). In the HP group, the percentage was higher than in the HF group for women only (*p* = 0.013) (Figure 3).

## 4. Discussion

In this study, the associations between dietary carbohydrate, protein, and fat intakes and the prevalence of MetS were investigated in Korean adults. The percentage of participants with MetS was significantly higher in the HCHO group than in the normal diet group in women. Comparisons with the normal diet group showed that an HCHO diet was positively associated with elevated blood pressure and triglyceride levels in women, while an HF diet in men was negatively associated with elevated fasting glucose levels.

Previous studies reported mixed findings regarding the association between dietary macronutrient intakes and the risk of MetS [25,27,28,29,30]. In a nationwide cross-sectional study, the risk of MetS in men increased in proportion to carbohydrate intake, regardless of the percentage of fat intake [30]. However, few studies have compared HCHO, HF, and HP diets with respect to the associations of dietary patterns with MetS. In the present study, the proportion of participants with MetS was considerably higher in the HCHO group than in the normal diet group in women. A meta-analysis of studies conducted in Asian countries indicated a positive association between carbohydrate consumption and MetS [37]. Several studies have shown that the metabolic impact of carbohydrates and fat depend more on their quality than quantity. Refined carbohydrates, such as those in cakes, syrups, and cookies, are closely related to MetS [38]. Saturated and trans fatty acids also have detrimental effects on metabolic health [39,40] whereas unsaturated fatty acids, including monounsaturated fatty acids and polyunsaturated fatty acids, improve metabolic markers [39,40].

Our study revealed positive associations of HCHO diet with elevated blood pressure and triglyceride levels in women based on a comparison with the normal diet group. These findings were similar to those in previous studies demonstrating that HCHO diets are related to MetS and its components [30,41,42,43,44]. In Korean women, a higher intake of carbohydrates was associated with a higher risk of MetS components, especially abdominal obesity [43]. In an observational study, a low-carbohydrate diet was associated with a significant reduction of blood pressure in patients with type 2 diabetes and glucose intolerance [45]. A study of US adults found that higher carbohydrate intake was associated with increased serum triglyceride and fasting blood glucose levels in women [44]. In a cross-sectional study of 245 elderly women, those with MetS had higher carbohydrate and lower protein intakes than those who did not. Positive and negative associations of carbohydrate intake with triglyceride and HDL-cholesterol levels, respectively, were also reported [42], as was an association between higher carbohydrate intake and unfavorable lipid metabolism, including increased triglyceride and decreased HDL-cholesterol levels [41]. Excessive intake of carbohydrates increases fructose-mediated triglyceride production [42]. In adipocytes, fructose serves as a substrate for lipogenesis, which generates new lipid vesicles and induces hepatic synthesis of triglyceride [46]. The prevalence of MetS increases with age but differs according to sex, dietary habits, socioeconomic status, and work-related activities [41,43]. In our study, an HF diet in men was associated with lower fasting glucose than the normal diet group. In a randomized trial of patients with type 2 diabetes, low-carbohydrate HF diets reduced fasting glucose and HbA1c levels [47]. An HF diet rich in unsaturated fatty acids, such as the Mediterranean diet, may prevent MetS, type 2 diabetes, and CVD [25]. The Mediterranean diet, which includes high amounts of olive oil, whole grains, nuts, seeds, fruit, and vegetables and low amounts of processed meats and sweets, can reduce the risk of MetS, type 2 diabetes, and CVD [48,49,50].

This study is the first to investigate the associations between macronutrient intake patterns (i.e., HCHO, HF, and HP diets) and MetS in the South Korean population according to sex using nationally representative data. However, several limitations of this study should be noted. First, it was difficult to establish the causality of the relationship between diet and MetS because of the cross-sectional design of the KNHANES. Second, the KNHANES does not provide information about the quality of dietary carbohydrates, fats, and proteins consumed—for example, in terms of glycemic load or the proportions of sugar, unsaturated/saturated fatty acids, and amino acids. Dietary glycemic load and glycemic index were developed to estimate insulinemia and postprandial glycemia, and they have been utilized to predict cardiometabolic risk factors [51]. A cross-sectional study showed that a higher dietary glycemic load and glycemic index were associated with an increased risk of MetS [52]. Several intervention studies suggested that exchanging dietary saturated fatty acids for monounsaturated fatty acids reduced LDL-cholesterol and triglyceride [53,54,55,56] and lowered blood pressure [54,57]. It would have been more helpful to have sufficient information about the quality of dietary macronutrients consumed if it could be provided. Third, intakes were assessed based on 24-h dietary recall data, where the dietary data might not have been representative of usual intakes because of day-to-day variation therein. Fourth, other factors that may affect metabolic risk factors, such as dietary supplements including vitamins and minerals, were not considered because of insufficient data.

## 5. Conclusions

An HCHO diet was strongly associated with a higher risk of MetS components in women, especially elevated blood pressure and high triglyceride levels; in men, an HF diet was negatively associated with elevated fasting glucose. A balanced diet with a recommended ratio of dietary macronutrients would be helpful for preventing MetS. Additional prospective investigations are warranted to determine the impact of dietary carbohydrate, fat, and protein proportions on metabolic health. The optimal types, and proportions of these dietary components through which suboptimal proportions can lead to MetS, should also be determined.

## Figures and Tables

**Figure 1 nutrients-15-02676-f001:**
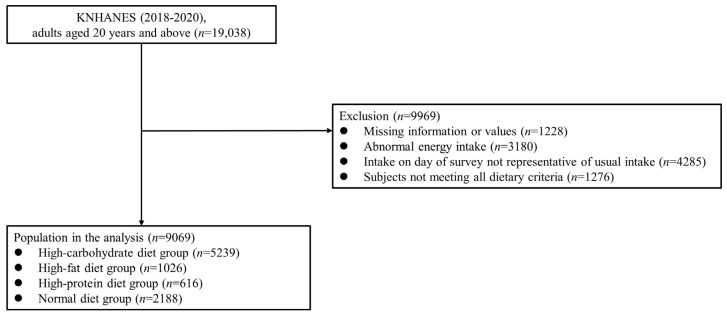
Study population: Data from the 2018–2020 Korean National Health and Nutrition Examination Survey (KNHANES).

**Figure 2 nutrients-15-02676-f002:**
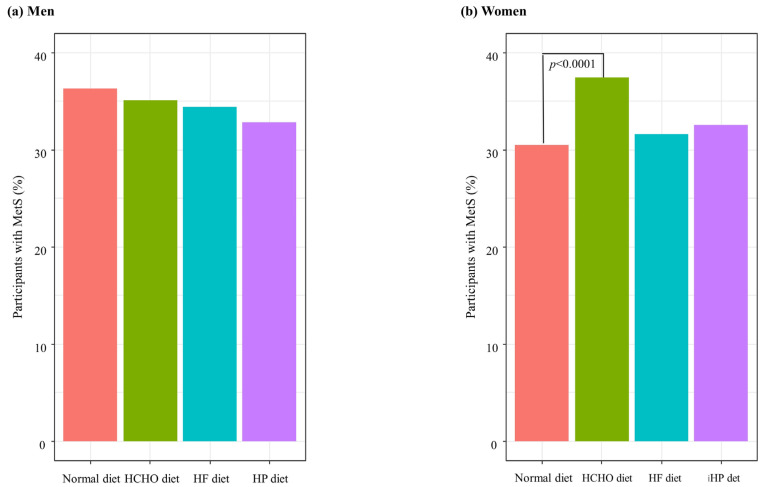
Percentages of participants with metabolic syndrome (MetS) according to dietary patterns. HCHO, high-carbohydrate; HF, high-fat; HP, high-protein. (**a**) In men, there were no significant differences of percentage of MetS between each group. (**b**) In women, the percentage of participants with MetS was significantly higher in the HCHO diet group than in the normal diet group (*p* < 0.0001).

**Figure 3 nutrients-15-02676-f003:**
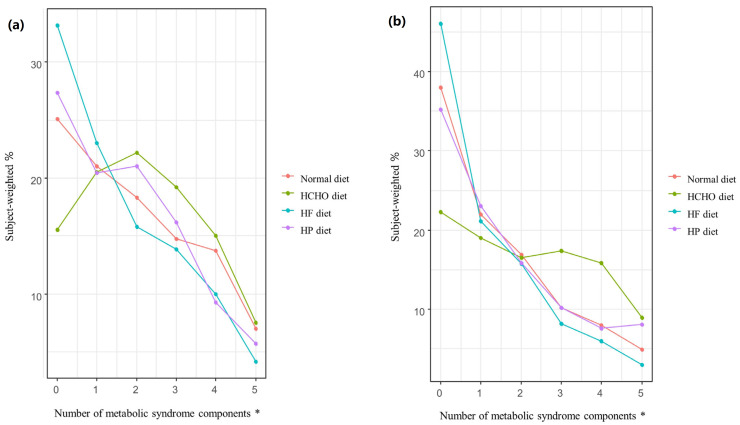
Percentage of participants in each diet group with metabolic syndrome according to the number of metabolic syndrome components. (**a**) In men, the rate of metabolic syndrome (MetS) in the high-carbohydrate (HCHO) group was higher than in the normal diet, high-fat (HF), and high-protein (HP) groups (*p* < 0.001), while the rate in the HF group was lower than in the normal diet group (*p* = 0.029) as the number of MetS components increased. (**b**) In women, the rate of MetS in the HCHO group was higher than in the normal diet, HF, and HP groups (*p* < 0.001), while the rate in the HP group was higher than in the HF group (*p* = 0.013). The rate of MetS in the HF group was lower than in the normal diet group (*p* = 0.033) as the number of MetS components increased. HCHO, high-carbohydrate; HF, high-fat; HP, high-protein; MetS, metabolic syndrome. * *p* value < 0.05.

**Table 1 nutrients-15-02676-t001:** Characteristics of the participants according to dietary pattern.

	Total	Men					Women				
		Normal Diet	HCHO Diet	HF Diet	HP Diet	*p*-Value	Normal Diet	HCHO Diet	HF Diet	HP Diet	*p*-Value
N	9069	980	2083	408	306	-	1208	3156	618	310	-
Age (y)	51.01 ± 0.31	46.38 ± 0.60	55.77 ± 0.50	39.98 ± 0.70	43.14 ± 0.95	<0.001	47.60 ± 0.56	57.36 ± 0.41	42.54 ± 0.69	46.24 ± 1.04	<0.001
Current smoking (%)	17.47	33.59	27.44	35.61	33.22	0.003	4.17	4.39	5.88	10.03	0.005
Heavy drinking (%)	8.27	13.54	10.83	17.91	17.64	0.001	4.04	2.67	4.65	7.61	0.001
Aerobic activity adherence (%)	42.85	47.38	41.31	54.56	54.7	<0.001	42.11	36.68	43.49	47.39	0.001
Household income (low, %)	25.13	21.78	29.08	20.75	22	<0.001	23.24	27.16	22.66	19.02	0.071
Energy intake (kcal/d)	1807.12 ± 10.18	2152.70 ± 23.85	1882.86 ± 17.77	2567.41 ± 47.99	2037.81 ± 47.97	<0.001	1612.73 ± 18.40	1478.18 ± 12.29	1861.66 ± 30.95	1475.10 ± 35.68	<0.001
Carbohydrate intake (g/d)	286.27 ± 1.45	326.16 ± 3.61	339.01 ± 3.04	288.04 ± 5.63	265.60 ± 6.46	<0.001	244.95 ± 2.81	271.31 ± 2.18	222.33 ± 4.09	194.37 ± 4.57	<0.001
Fat intake (g/d)	41.69 ± 0.47	55.07 ± 0.77	29.29 ± 0.47	103.75 ± 2.23	46.18 ± 1.42	<0.001	42.58 ± 0.57	22.58 ± 0.31	75.28 ± 1.32	35.51 ± 1.07	<0.001
Protein intake (g/d)	65.42 ± 0.48	79.94 ± 0.92	60.77 ± 0.65	93.03 ± 1.82	119.81 ± 2.87	<0.001	59.74 ± 0.70	46.54 ± 0.47	67.42 ± 1.26	78.03 ± 2.19	<0.001
Fiber intake (g/d)	24.76 ± 0.19	26.84 ± 0.46	27.65 ± 0.36	24.59 ± 0.63	25.17 ± 0.85	0.546	21.68 ± 0.36	24.14 ± 0.32	21.02 ± 0.59	20.46 ± 0.64	0.023
Energy from											
Carbohydrate (%)	64.67 ± 0.18	60.67 ± 0.10	72.44 ± 0.15	45.77 ± 0.47	52.66 ± 0.48	<0.001	60.80 ± 0.09	73.75 ± 0.14	47.94 ± 0.34	53.01 ± 0.49	<0.001
Fat (%)	19.76 ± 0.14	22.88 ± 0.13	13.64 ± 0.13	36.13 ± 0.30	20.16 ± 0.34	<0.001	23.60 ± 0.12	13.43 ± 0.12	36.37 ± 0.25	21.57 ± 0.33	<0.001
Protein (%)	14.42 ± 0.06	14.94 ± 0.10	12.86 ± 0.07	14.51 ± 0.14	23.64 ± 0.24	<0.001	14.93 ± 0.08	12.55 ± 0.06	14.55 ± 0.13	23.80 ± 0.28	<0.001
Comorbidity (%)											
Diabetes	14.13	14.73	18.89	8.75	11.25	<0.001	8.39	16.79	6.5	8.9	<0.001
Hypertension	30.61	29.23	37.14	20.93	21.51	<0.001	20.89	39.21	14.26	21.46	<0.001
Dyslipidemia	15.97	12.67	16.21	7.19	12.27	<0.001	13.28	22.97	9.6	14.97	<0.001
Waist circumference (cm)	83.41 ± 0.14	86.85 ± 0.35	87.37 ± 0.25	86.65 ± 0.49	86.93 ± 0.62	0.162	78.77 ± 0.36	81.43 ± 0.23	76.62 ± 0.50	79.13 ± 0.64	0.002
Body mass index (kg/m^2^)	23.88 ± 0.05	24.50 ± 0.14	24.29 ± 0.09	24.50 ± 0.19	24.81 ± 0.23	0.230	23.10 ± 0.15	23.69 ± 0.08	22.52 ± 0.19	23.35 ± 0.23	0.028
SBP (mmHg)	118.79 ± 0.25	119.41 ± 0.50	121.89 ± 0.40	116.70 ± 0.73	117.84 ± 0.86	0.260	114.38 ± 0.51	121.26 ± 0.45	110.86 ± 0.69	112.81 ± 0.97	<0.001
DBP (mmHg)	75.48 ± 0.15	78.35 ± 0.37	76.49 ± 0.28	77.63 ± 0.54	78.25 ± 0.59	0.286	73.48 ± 0.31	74.17 ± 0.22	72.33 ± 0.45	74.09 ± 0.61	0.122
Fasting glucose (mg/dL)	101.22 ± 0.30	103.05 ± 0.98	105.78 ± 0.67	98.29 ± 1.10	99.36 ± 1.02	0.701	96.15 ± 0.56	102.16 ± 0.55	94.35 ± 0.73	95.96 ± 0.91	0.491
Total cholesterol (mg/dL)	190.66 ± 0.50	190.32 ± 1.35	186.42 ± 1.01	192.89 ± 1.89	190.96 ± 2.46	0.425	193.22 ± 1.18	192.00 ± 0.81	192.23 ± 1.51	192.77 ± 2.36	0.603
LDL-cholesterol (mg/dL)	116.12 ± 1.15	120.52 ± 3.08	112.05 ± 1.92	116.17 ± 4.21	120.82 ± 5.07	0.222	125.20 ± 3.78	115.36 ± 2.24	120.97 ± 5.97	105.01 ± 6.68	0.004
HDL-cholesterol (mg/dL)	51.16 ± 0.16	47.67 ± 0.41	46.45 ± 0.31	48.05 ± 0.54	48.64 ± 0.77	0.441	56.32 ± 0.44	53.28 ± 0.26	58.05 ± 0.64	55.70 ± 0.88	0.324
Triglycerides (mg/dL)	131.55 ± 1.42	148.73 ± 4.27	154.93 ± 3.50	150.11 ± 7.10	140.95 ± 6.00	0.921	104.74 ± 1.91	120.86 ± 1.84	98.01 ± 2.35	112.14 ± 6.77	0.014

Values are means ± standard errors or percentages. HCHO, high-carbohydrate; HF, high-fat; HP, high-protein; SBP, systolic blood pressure; DBP, diastolic blood pressure; LDL, low-density lipoprotein; HDL, high-density lipoprotein.

**Table 2 nutrients-15-02676-t002:** Associations between metabolic syndrome components and dietary pattern according to sex.

	Unadjusted	*p* Value	Model 1	*p* Value	Model 2	*p* Value	Model 3	*p* Value
Men								
Increased waist circumference								
Normal diet	1	-	1	-	1	-	1	-
HCHO diet	0.988 (0.827–1.181)	0.895	0.862 (0.715–1.039)	0.119	0.740 (0.542–1.010)	0.058	0.813 (0.606–1.089)	0.165
HF diet	0.850 (0.649–1.113)	0.237	0.944 (0.720–1.237)	0.675	0.872 (0.551–1.380)	0.558	0.880 (0.565–1.370)	0.570
HP diet	0.890 (0.644–1.232)	0.483	0.945 (0.683–1.308)	0.735	0.675 (0.389–1.173)	0.163	0.751 (0.432–1.305)	0.308
Elevated blood pressure								
Normal diet	1	-	1	-	1	-	1	-
HCHO diet	*1.303 (1.101–1.543)*	*0.002*	0.893 (0.740–1.077)	0.236	0.855 (0.697–1.048)	0.132	0.845 (0.693–1.031)	0.096
HF diet	*0.632 (0.486–0.823)*	*0.001*	0.833 (0.630–1.103)	0.201	0.822 (0.602–1.122)	0.216	0.863 (0.631–1.180)	0.354
HP diet	0.807 (0.592–1.100)	0.174	0.947 (0.680–1.318)	0.745	0.854 (0.597–1.221)	0.387	0.845 (0.587–1.217)	0.364
Elevated fasting glucose								
Normal diet	1	-	1	-	1	-	1	-
HCHO diet	*1.338 (1.114–1.607)*	*0.002*	0.956 (0.785–1.165)	0.658	0.978 (0.789–1.212)	0.836	0.982 (0.791–1.219)	0.868
HF diet	*0.579 (0.433–0.774)*	*<0.001*	*0.737 (0.549–0.991)*	*0.044*	*0.686 (0.498–0.945)*	*0.021*	*0.671 (0.488–0.922)*	*0.014*
HP diet	0.752 (0.555–1.020)	0.067	0.863 (0.632–1.180)	0.356	0.800 (0.571–1.122)	0.196	0.727 (0.518–1.020)	0.065
Elevated triglycerides								
Normal diet	1	-	1	-	1	-	1	-
HCHO diet	1.055 (0.888–1.252)	0.542	0.937 (0.785–1.117)	0.465	0.976 (0.802–1.186)	0.804	1.009 (0.831–1.226)	0.925
HF diet	0.856 (0.656–1.118)	0.253	0.938 (0.721–1.220)	0.632	0.914 (0.689–1.210)	0.528	0.912 (0.679–1.224)	0.537
HP diet	0.871 (0.637–1.192)	0.388	0.918 (0.672–1.253)	0.588	0.887 (0.632–1.247)	0.490	0.886 (0.641–1.225)	0.464
Reduced HDL-cholesterol								
Normal diet	1	-	1	-	1	-	1	-
HCHO diet	*1.380 (1.136–1.677)*	*0.001*	1.100 (0.899–1.345)	0.354	1.042 (0.841–1.291)	0.708	1.110 (0.896–1.376)	0.339
HF diet	0.841 (0.630–1.121)	0.237	1.015 (0.758–1.359)	0.921	1.068 (0.781–1.460)	0.679	1.121 (0.821–1.530)	0.471
HP diet	1.089 (0.797–1.487)	0.594	1.221 (0.892–1.673)	0.213	1.257 (0.897–1.762)	0.183	1.362 (0.982–1.888)	0.064
Women								
Increased waist circumference								
Normal diet	1	-	1	-	1	-	1	-
HCHO diet	*1.614 (1.351–1.927)*	*<0.001*	1.117 (0.925–1.349)	0.249	0.928 (0.682–1.261)	0.631	1.018 (0.752–1.378)	0.909
HF diet	0.761 (0.572–1.013)	0.061	0.946 (0.702–1.275)	0.715	0.973 (0.585–1.618)	0.915	1.037 (0.640–1.681)	0.882
HP diet	1.021 (0.737–1.413)	0.902	1.131 (0.817–1.564)	0.457	0.968 (0.570–1.644)	0.904	1.034 (0.596–1.797)	0.904
Elevated blood pressure								
Normal diet	1	-	1	-	1	-	1	-
HCHO diet	*2.213 (1.873–2.615)*	*<0.001*	*1.245 (1.030–1.505)*	*0.024*	*1.233 (1.007–1.510)*	*0.043*	*1.257 (1.021–1.548)*	*0.032*
HF diet	*0.674 (0.521–0.870)*	*0.003*	0.978 (0.713–1.340)	0.888	1.040 (0.737–1.466)	0.824	1.073 (0.749–1.535)	0.701
HP diet	1.027 (0.740–1.426)	0.872	1.293 (0.889–1.882)	0.179	1.267 (0.863–1.861)	0.227	1.274 (0.868–1.871)	0.216
Elevated fasting glucose								
Normal diet	1	-	1	-	1	-	1	-
HCHO diet	*1.822 (1.523–2.179)*	*<0.001*	*1.210 (1.001–1.464)*	*0.049*	1.143 (0.938–1.393)	0.186	1.092 (0.885–1.347)	0.411
HF diet	*0.754 (0.577–0.986)*	*0.039*	0.969 (0.732–1.283)	0.827	1.015 (0.752–1.370)	0.922	0.980 (0.729–1.318)	0.895
HP diet	1.027 (0.743–1.419)	0.873	1.161 (0.840–1.606)	0.366	1.042 (0.736–1.476)	0.815	1.020 (0.725–1.437)	0.908
Elevated triglycerides								
Normal diet	1	-	1	-	1	-	1	-
HCHO diet	*1.902 (1.609–2.248)*	*<0.001*	*1.317 (1.101–1.577)*	*0.003*	*1.339 (1.098–1.632)*	*0.004*	*1.329 (1.091–1.619)*	*0.005*
HF diet	0.871 (0.664–1.143)	0.319	1.108 (0.829–1.480)	0.487	1.141 (0.832–1.566)	0.413	1.054 (0.770–1.442)	0.743
HP diet	1.223 (0.888–1.686)	0.217	*1.388 (1.009–1.910)*	*0.044*	1.317 (0.941–1.843)	0.108	1.222 (0.858–1.741)	0.266
Reduced HDL-cholesterol								
Normal diet	1	-	1	-	1	-	1	-
HCHO diet	*1.678 (1.432–1.966)*	*<0.001*	*1.191 (1.003–1.415)*	*0.046*	1.183 (0.990–1.415)	0.064	1.170 (0.972–1.409)	0.097
HF diet	*0.721 (0.568–0.917)*	*0.008*	0.876 (0.681–1.128)	0.304	0.871 (0.666–1.138)	0.311	0.877 (0.670–1.147)	0.336
HP diet	1.178 (0.859–1.615)	0.308	1.311 (0.944–1.822)	0.106	1.272 (0.913–1.771)	0.155	1.214 (0.863–1.707)	0.265

Values are expressed as odds ratios (with 95 % confidence intervals). HCHO, high-carbohydrate; HF, high-fat; HP, high-protein; HDL, high-density lipoprotein. Model 1: with adjustment for age. Model 2: Model 1 with adjustment for body mass index, smoking status, alcohol consumption, aerobic activity adherence, household income. Model 3: Model 2 with adjustment for daily energy and fiber intake. Results in italics indicate statistical significance at the 0.05 level.

## Data Availability

The data are available at https://knhanes.cdc.go.kr/knhanes/eng/index.do (accessed on 14 April 2022).

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
