# Peer review of "Associations between Dietary Patterns and Metabolic Syndrome: Findings of the Korean National Health and Nutrition Examination Survey"

_nutrients, 2023, doi:10.3390/nu15122676_

Round 1

Reviewer 1 Report

The manuscript by Lee et al. conveyed an investigation on the associations between dietary patterns and metabolic syndromes based on the Korean National Health and Nutrition Examination Survey. This study mainly reported the findings that high carbohydrate diet was strongly associated with a higher risk of metabolic syndrome (MetS) components in women, and a high fat diet was negatively associated with MetS component(s) in men in the South Korean population, with specific analytical models that were adjusted to variables regarding the age, sex, smoking status, alcohol consumption, physical activity, and household income, etc. Overall, I think this work serves the readers well with the meaningful information regarding the previously ambiguous findings of the association between dietary macronutrient intake and the risk of metabolic syndromes with rather recent (2018-2020) and thoroughly collected data (nation-wide survey). There are a few points that requires improvements or clarification that will help to improve the quality of this article:

Major point:

In the “3.3 Associations between metabolic syndrome components and dietary patterns” section of the Results, the authors described in details about how HCHO or HF are significantly associated with the different components of MetS. Yet not all of these conclusions were made with a specific note on which model the result is based on. For example, HCHO diet was significantly associated with elevated blood pressure in men according to “Unadjusted”, but not Model 1 through 3. The authors should bring the readers’ attention to the facts that which conclusions are made with/without certain adjustment/model. A figure or table to specifically highlight these significant findings might also be helpful to specify this point to provide thorough information according the nuance of analytical adjustments since the variables are quite important factors relating to MetS.

Minor points:

1.     The lower halves, including the X-axes of Figure 1 and Figure 2 are missing due to formatting issues. This does not affect my understanding based on the descriptions in the Results section. As the authors provide the intact images for these figures, I leave it to the editor’s discretion (since I voted for the “Accept after minor changes” and might not see the final version of this manuscript before publication) to make sure the figures match the description and conclusions made by the authors.

2.     Throughout the manuscript, the authors used the term “Control” to describe the group with normal diet, as in contrast with HCHO, HF, and HP. I suggest to change the wording of “Control” to “Normal Diet Group”, due to the fact that this is an observational study rather than a performed experiment/trial and the word “Control” might be misleading and not describe the “Normal Diet Group” properly.

3. The authors thoroughly discussed the limitations of this study, with the second point about that the KNHANES does not provide information about the quality of the macronutrients. The authors could elaborate about how this might impact the findings of this study by relating back to the points made in the second paragraph of the Discussion section on the confounding effects of the different quality of the macronutrients. This will help strengthen the logic flow of the manuscript.

Author Response

We thank the reviewers for their valuable opinions and sincere and detailed comments. We submit a revised version of the manuscript that includes modifications and amendments based on the reviewers’ opinions.

The followings are responses to each of the reviewers’ comments:

# Reviewer 1.

Comments and Suggestions for Authors:

The manuscript by Lee et al. conveyed an investigation on the associations between dietary patterns and metabolic syndromes based on the Korean National Health and Nutrition Examination Survey. This study mainly reported the findings that high carbohydrate diet was strongly associated with a higher risk of metabolic syndrome (MetS) components in women, and a high fat diet was negatively associated with MetS component(s) in men in the South Korean population, with specific analytical models that were adjusted to variables regarding the age, sex, smoking status, alcohol consumption, physical activity, and household income, etc. Overall, I think this work serves the readers well with the meaningful information regarding the previously ambiguous findings of the association between dietary macronutrient intake and the risk of metabolic syndromes with rather recent (2018-2020) and thoroughly collected data (nation-wide survey). There are a few points that requires improvements or clarification that will help to improve the quality of this article:

Major point:

In the “3.3 Associations between metabolic syndrome components and dietary patterns” section of the Results, the authors described in details about how HCHO or HF are significantly associated with the different components of MetS. Yet not all of these conclusions were made with a specific note on which model the result is based on. For example, HCHO diet was significantly associated with elevated blood pressure in men according to “Unadjusted”, but not Model 1 through 3. The authors should bring the readers’ attention to the facts that which conclusions are made with/without certain adjustment/model. A figure or table to specifically highlight these significant findings might also be helpful to specify this point to provide thorough information according the nuance of analytical adjustments since the variables are quite important factors relating to MetS.

Answer) Thank you for your detailed and valuable opinions. We revised the manuscript according to your comments, described in detail, and summarized the results at a glance, focusing on the contents that showed significant findings in model 1 through 3 below. To specifically highlight these important results, significant results are highlighted in italics in Table 2.

3.3. Associations between metabolic syndrome components and dietary patterns

The unadjusted odds ratios (ORs), age-adjusted ORs (model 1), and multivariate-adjusted ORs (models 2 and 3) of MetS components according to dietary patterns and sex are shown in Table 2. In men, an HF diet was negatively associated with elevated fasting glucose levels based on a comparison with the normal diet group (OR = 0.579, 95% CI: 0.433–0.774, p<0.001). This negative association remained significant after adjusting for age, BMI, smoking status, alcohol consumption, aerobic activity adherence, household income, daily energy intake, and fiber intake (OR = 0.671, 95% CI: 0.488–0.922, p=0.014). In women, an HCHO diet was positively associated with elevated blood pressure and triglycerides levels based on a comparison with the normal diet group (OR = 2.213, 95% CI: 1.873–2.615, p<0.001; OR = 1.902, 95% CI: 1.609–2.248, p<0.001, respectively). These associations remained significant after adjusting for age, BMI, smoking status, alcohol consumption, aerobic activity adherence, household income, daily energy intake, and fiber intake (OR = 1.257, 95% CI: 1.021–1.548, p=0.032; OR = 1.329, 95% CI: 1.091–1.619, p=0.005, respectively) (Table 2).

Minor points:

  1. The lower halves, including the X-axes of Figure 1 and Figure 2 are missing due to formatting issues. This does not affect my understanding based on the descriptions in the Results section. As the authors provide the intact images for these figures, I leave it to the editor’s discretion (since I voted for the “Accept after minor changes” and might not see the final version of this manuscript before publication) to make sure the figures match the description and conclusions made by the authors.

Answer) Thank you so much for your sincere and considerate comments. We submitted the intact figures, but there seems to be an error in the formatting process. As you mentioned, we checked the correct figures and inserted them into the manuscript so that they would not be missed.

  1. Throughout the manuscript, the authors used the term “Control” to describe the group with normal diet, as in contrast with HCHO, HF, and HP. I suggest to change the wording of “Control” to “Normal Diet Group”, due to the fact that this is an observational study rather than a performed experiment/trial and the word “Control” might be misleading and not describe the “Normal Diet Group” properly.

Answer) Thank you for your detailed and sincere comments. As you suggested, we changed the wording of “Control” to “Normal Diet Group” in the manuscript. The “Normal Diet Group” is considered to be an appropriate expression. Thank you for your valuable opinions.

  1. The authors thoroughly discussed the limitations of this study, with the second point about that the KNHANES does not provide information about the quality of the macronutrients. The authors could elaborate about how this might impact the findings of this study by relating back to the points made in the second paragraph of the Discussion section on the confounding effects of the different quality of the macronutrients. This will help strengthen the logic flow of the manuscript.

Answer) Thank you so much for your valuable suggestions. We revised the manuscript according to your comments and described it in the Discussion section below.

“Second, the KNHANES does not provide information about the quality of dietary carbohydrates, fats, and proteins consumed, for example in terms of glycemic load or the proportions of sugar, unsaturated/saturated fatty acids, and amino acids. Dietary glycemic load and glycemic index were developed to estimate insulinemia and postprandial glycemia. And they have been utilized to predict cardiometabolic risk factors [42]. A cross-sectional study showed that higher dietary glycemic load and glycemic index were associated with increased risk of MetS [43]. Several intervention studies suggested that exchanging dietary saturated fatty acids for monounsaturated fatty acids reduced LDL-cholesterol and triglyceride [44-47], and lowered blood pressure [45,48]. If sufficient information about the quality of dietary macronutrients consumed could be provided, it would have been more helpful.”

Reviewer 2 Report

An interesting study considering that the metabolic syndrome is a complex of imbalances in the body and represents a risk factor for complications with risk of death. Diet and lifestyle play an important role in the prevention and improvement of metabolic syndrome. In order to publish the study, a series of changes are necessary:

- first of all, the introduction is very brief, the prevalence of the metabolic syndrome, especially in young people, should have been documented much more and a much more detailed highlighting of the link between the quality of food and the metabolic syndrome, as well as its complications.

- a more detailed specification of the method of carrying out the study, the design of the questionnaire used, its validation or if a validated questionnaire was used is also necessary.

- related to data processing and the presentation of information about the study respondents, a presentation of their classification according to BMI groups would have been interesting.

- The conclusions are also brief, a clearer detailing of the importance of the study is needed and a report of it to similar studies specifying the novelty brought. Also, nothing is specified about the limits of the study. It would be preferable to formulate a series of suggestions regarding the correction of nutritional behaviour for preventive or balancing purposes.

Author Response

We thank the reviewers for their valuable opinions and sincere and detailed comments. We submit a revised version of the manuscript that includes modifications and amendments based on the reviewers’ opinions.

The followings are responses to each of the reviewers’ comments:

# Reviewer 2.

Comments and Suggestions for Authors:

An interesting study considering that the metabolic syndrome is a complex of imbalances in the body and represents a risk factor for complications with risk of death. Diet and lifestyle play an important role in the prevention and improvement of metabolic syndrome. In order to publish the study, a series of changes are necessary:

- first of all, the introduction is very brief, the prevalence of the metabolic syndrome, especially in young people, should have been documented much more and a much more detailed highlighting of the link between the quality of food and the metabolic syndrome, as well as its complications.

Answer) Thank you for your considerate and sincere comments. As you mentioned, the details related to the prevalence of metabolic syndrome, especially in young people, and the link between the quality of food and the metabolic syndrome have been noted in the Introduction section and highlighted.

- a more detailed specification of the method of carrying out the study, the design of the questionnaire used, its validation or if a validated questionnaire was used is also necessary.

Answer) Thank you so much for your sincere and detailed comments and valuable opinions. We revised the manuscript according to your comments and some references have been added to the Method section below.

"Data are collected via household interviews and anthropometric, biochemical, and nutritional status assessments [30]. ~ Dietary intake was estimated from the food composition tables of the Rural Development Administration, in combination with the nutrient database of the Korea Health and Industry of Development Institute [31,32]."

  1. Korean Centers for Disease Control and Prevention. Korea National Health and Nutrition Examination Survey. Available online: https://knhanes.kdca.go.kr/.
  2. Paik, H.Y. Dietary reference intakes for Koreans (KDRIs). 2008, 17, 416-419.
  3. Dietary reference intakes for Koreans 2020. The Korean Nutrition Society: 2020.

- related to data processing and the presentation of information about the study respondents, a presentation of their classification according to BMI groups would have been interesting.

Answer) Thank you for your considerate and valuable opinions. We described the differences in BMI according to dietary patterns in Table 1. According to the reviewer’s valuable comments, it would be useful and interesting if additional studies on obesity and body composition such as BMI and body fat according to dietary patterns are conducted. Thank you so much for your valuable suggestions.

- The conclusions are also brief, a clearer detailing of the importance of the study is needed and a report of it to similar studies specifying the novelty brought. Also, nothing is specified about the limits of the study. It would be preferable to formulate a series of suggestions regarding the correction of nutritional behaviour for preventive or balancing purposes.

Answer) Thank you for your sincere and detailed suggestions and valuable opinions. This study is the first to investigate the associations between macronutrient intake patterns (high-carbohydrate [HCHO], high-fat [HF], and high-protein [HP] diets) and MetS in the South Korean population according to sex using nationally representative data (KNHANES 2018-2020). As you mentioned, we revised the manuscript and the details have been noted in the Discussion section and highlighted.

Round 2

Reviewer 2 Report

Please highlight the conclusions in a separate subchapter.